

# The relationship between the female athlete triad and injury rates in collegiate female athletes

Mutsuaki Edama[1,2], Hiromi Inaba[1,3], Fumi Hoshino[3], Saya Natsui[3], Sae Maruyama[2] and Go Omori[1,4]

[1] Athlete Support Research Center, Niigata University of Health and Welfare, Niigata, Japan, Niigata, Niigata, Japan
[2] Institute for Human Movement and Medical Sciences, Niigata University of Health and Welfare, Niigata, Japan, Niigata, Niigata, Japan
[3] Department of Health and Nutrition, Niigata University of Health and Welfare, Niigata, Japan, Niigata, Japan
[4] Department of Health and Sports, Niigata University of Health and Welfare, Niigata, Japan, Niigata, Niigata, Japan

Corresponding author
Mutsuaki Edama,
edama@nuhw.ac.jp

## ABSTRACT

**Background:** This study aimed to clarify the relationship between the triad risk assessment score and the sports injury rate in 116 female college athletes (average age, 19.8 ± 1.3 years) in seven sports at the national level of competition; 67 were teenagers, and 49 were in their 20s.

**Methods:** Those with menstrual deficiency for >3 months or <6 menses in 12 months were classified as amenorrheic athletes. Low energy availability was defined as adolescent athletes having a body weight <85% of ideal body weight, and for adult athletes in their 20s, a body mass index ≤17.5 kg/m². Bone mineral density (BMD) was measured on the heel of the right leg using an ultrasonic bone densitometer. Low BMD was defined as a BMD Z-score <−1.0. The total score for each athlete was calculated. The cumulative risk assessment was defined as follows: low risk (a total score of 0–1), moderate risk (2–5), and high risk (6). The injury survey recorded injuries referring to the injury survey items used by the International Olympic Committee.

**Results:** In swimming, significantly more athletes were in the low-risk category than in the moderate and high-risk categories ($p = 0.004$). In long-distance athletics, significantly more athletes were in the moderate-risk category than in the low and high-risk categories ($p = 0.004$). In the moderate and high-risk categories, significantly more athletes were in the injury group, whereas significantly more athletes in the low-risk category were in the non-injury group ($p = 0.01$). Significantly more athletes at moderate and high-risk categories had bone stress fractures and bursitis than athletes at low risk ($p = 0.023$).

**Discussion:** These results suggest that athletes with relative energy deficiency may have an increased injury risk.

## INTRODUCTION

The female athlete triad (hereafter referred to as triad) has three components: (1) low energy availability (LEA) with or without disordered eating (DE)/eating disordered (ED); (2) menstrual dysfunction; and (3) low bone mineral density (BMD) (*De Souza et al., 2014*). An IOC consensus group has recently introduced a new umbrella term, that is, 'Relative Energy Deficiency in Sport' (RED-S), to describe the physiological and pathophysiological effects of energy deficiency in male and female athletes (*Mountjoy et al., 2014*). The authors assert that "RED-S is required to more accurately describe the clinical syndrome originally known as the Female Athlete Triad" that is a "more comprehensive, broader term for the overall syndrome, which includes what has so far been called the "Female Athlete Triad" (*Mountjoy et al., 2014*)." RED-S is based on a relative energy deficit that is reported to affect various factors (*Mountjoy et al., 2018*). However, RED-S is insufficiently supported by scientific research to warrant adoption at this time. Recently, there have been many studies of bone stress fractures and amenorrhea, but their relationships with the occurrence of sports injury have not been examined (*De Souza et al., 2014*; *Nattiv, 2000*; *Nose-Ogura et al., 2019*; *Reeder et al., 1996*; *Takamatsu & Kitawaki, 2016*). It was previously reported that the frequency of sports injuries was higher in women than in men, suggesting a relationship between the menstrual cycle and sports injuries (*Hewett, Zazulak & Myer, 2007*; *Park et al., 2009*). Especially for ACL injury, many studies have been reported. Gender differences in the incidence of ACL injury include extrinsic factors (physical and visual perturbations, bracing, and shoe-surface interaction) and intrinsic factors (anatomical, neuromuscular, and biomechanical differences between genders), which may be multifactorial in nature (*Hewett, Myer & Ford, 2006*). Among the intrinsic factors, the effects of female hormones are also thought to play a role in gender differences in the incidence of ACL injury (*Arendt, Bershadsky & Agel, 2002*; *Beynnon et al., 2006*; *Hewett, Myer & Ford, 2006*; *Wojtys et al., 2002*; *Wojtys et al., 1998*). It has been suggested that there is a strong relationship between the risk of both RED-S and sports injuries.

The advances in our understanding of risk factors and management of the triad are reflected in evidence based guidelines developed by the Female Athlete Triad Coalition in 2014 to help guide medical decision making for female athletes (*De Souza et al., 2014*). The resulting Female Athlete Triad Cumulative Risk Assessment includes the following 6 items scored on a scale from 0 to 2: low LEA with or without DE/ED; low body mass index (BMI); delayed menarche; oligomenorrhea or amenorrhea; low BMD; and prior stress fracture (*De Souza et al., 2014*). The resulting risk assessment score is used to classify an athlete into 1–3 categories: low risk (0–1 points), moderate risk (2–5 points), or high risk (six points) (*De Souza et al., 2014*).

Using risk assessment scores to help manage treatment for athletes is important, especially considering the evidence for adverse health consequences resulting from the triad. For example, a higher number of triad risk factors is associated with an increased risk for bone stress injuries and low BMD (*Barrack et al., 2014*; *Gibbs et al., 2014*; *Tenforde et al., 2013*). Furthermore, for female athletes who have one component of the triad, the

risk of developing bone stress fractures is about three times higher than that of athletes with no components of the triad; the risk is about five times higher for those with two or more components (*Mallinson & De Souza, 2014*). In addition, collegiate athletes with triad risk factors including oligomenorrhea/amenorrhea or increased risk assessment scores had higher grade bone stress fractures on MRI and longer return to play (*Nattiv et al., 2013*).

Therefore, this study aimed to clarify the relationship between the triad risk assessment score and the number of sports injuries. The hypothesis of this study was that the moderate and high-risk groups have higher injury rates than the low-risk group on the triad risk assessment score.

## MATERIALS AND METHODS

### Recruitment

A total of 116 female college athletes (average age, 19.8 ± 1.3 years) were investigated; 67 were teenagers, and 49 were in their 20s. They were involved in seven sports (swimming, athletics sprint, athletics long-distance, athletics throwing/jumping, soccer, basketball and volleyball). All sports were at the national level of competition. Approval was obtained from the Ethics Committee of The Niigata University of Health and Welfare to carry out this study within its facilities (approval no. 18032). Written, informed consent was obtained from all participants.

### Self-report using the questionnaire form

The examinations were conducted from August 2018 to January 2019. The participants were asked about age at menarche, date of last menstrual period, number of menstrual cycles per 12 months, history of bone stress fracture (site and times), dietary restriction, and present or past history of ED/DE using the questionnaire form. Those with menstrual deficiency for >3 months (definition of the Japan Society of Obstetrics and Gynecology) or <6 menses in 12 months were classed as amenorrheic athletes (*De Souza et al., 2014*). Data were collected by physical therapists (M.E.) and nutritionists (H.I. and F.H.).

### Anthropometry

Height (m) and body weight (kg) were measured using a body composition monitor (DC150; TANITA, Tokyo, Japan). BMD was measured on the heel of the right leg using an ultrasonic bone densitometer (AOS-100SA; Hitachi Aloka Medical, Tokyo, Japan). BMD was calculated as osteo-sonoassessment index (OSI) as follows: OSI = transmission index (TI) × speed of sound $(SOS)^2$, calculated as SOS (m/s) = heel width (m) × ultrasonic wave propagation time (sec). A BMD Z-score of < −1.0 in the heel is defined as low BMD (as defined by the Triad coalition in 2014). The LEA is defined as energy intake minus energy expenditure of exercise relative to fat-free mass (FFM) <30 kcal/kg of FFM/d, but it is very difficult to calculate energy balance this way during medical examinations. Therefore, the American College of Sports Medicine defines LEA in adolescent athletes as a body weight <85% of ideal body weight (IBW), and for adult athletes in their 20s, a BMI ≤17.5 $kg/m^2$ (*De Souza et al., 2014*; *Nose-Ogura et al., 2019*).

Therefore, these criteria were used in the present study (*De Souza et al., 2014*). BMI was calculated as body weight (kg)/height (m$^2$). To calculate IBW, the formula recommended by The Japanese Society for Pediatric Endocrinology was used.

## The female athlete triad cumulative risk assessment

The Female Athlete Triad Cumulative Risk Assessment was used. The following six factors were scored: (1) LEA with or without DE/ED; (2) low BMI; (3) delayed menarche; (4) oligomenorrhea and/or amenorrhea; (5) low BMD; and (6) stress reaction/fractures. With respect to LEA, athletes who received treatment by a psychiatrist received a score of 2, those with some dietary restriction as evidenced by self-report or low/inadequate energy intake on diet logs received a score of 1, and those with no history received a score of 0. BMI was scored for athletes over 20 years of age, but IBW was used for teenagers. Athletes with a BMI ≤17.5 kg/m$^2$ or IBW <85% received a score of 2, and athletes with a BMI between 17.6 and 18.4 kg/m$^2$ or IBW <90% received a score of 1. A score of 0 was given to athletes with a BMI ≥18.5 kg/m$^2$ or IBW ≥ 90%. For delayed menarche, athletes who had their menarche at age >16 years received a score of 2, athletes who had their menarche at age 15–16 years received a score of 1, and those with menarche at under 15 years received a score of 0. Athletes with amenorrhea (>3 months or <6 menses in 12 months) were scored 2, 6–9 menses in 12 months were scored 1, and eumenorrheic athletes (>9 menses in 12 months) were scored 0. For low BMD, athletes with a Z-score ≤−2 were scored 2, and those between −1 and −2 were scored 1; a score of 0 was given to those over −1. For a history of stress fractures, those with a history of 2 or more stress fractures or trabecular bone stress fractures were scored 2, those with only one past stress fracture were scored 1, and those with no stress fractures were scored 0. Next, the total score for each athlete was calculated, and the cumulative risk assessment was defined as follows: low risk (a total score of 0–1), moderate risk (a score of 2–5), and high risk (a score of 6) (*De Souza et al., 2014*).

## Number of injuries

An injury survey during sports activities was conducted for one season from April 2018 to March 2019. The injury survey collected injuries that resulted in failure to participate in practice and competition for more than 24 h after injury, referring to the injury survey items used by the IOC (*Junge et al., 2008*). Injured body part location and type of injury diagnosis were recorded. Data were collected by seven physical therapists and a medical doctor on the field and in the hospital. Serious illnesses such as stress fractures were diagnosed by a medical doctor at the hospital using X-rays and MRI.

## Statistical analysis

Pearson's chi-square test was used for comparisons of differences in the number of injuries by risk categories. Fisher's exact test was used for comparisons of differences in the risk categories for each sport, and to compare differences in injured body part-location and the type of injury diagnosis by risk category. Multiple comparisons were performed

using the Ryan nominal level for post hoc testing. Statistical analyses were performed using SPSS (Version 26.0; SPSS Japan Inc., Tokyo, Japan). The level of significance was $p < 0.05$.

## RESULTS

### Athletes' characteristics

The sports undertaken by the participants were swimming ($n = 11$), athletics sprint ($n = 19$), athletics long-distance ($n = 8$), athletics throwing/jumping ($n = 8$), soccer ($n = 27$), basketball ($n = 26$), and volleyball ($n = 17$).

### The three triad components

There were 4/116 (3.4%) athletes with LEA (defined as actual body weight of IBW <85% for adolescent athletes and a BMI ≤17.5 kg/m$^2$ for adult athletes), 6/116 (5.2%) athletes with amenorrhea (>3 months or <6 menses in 12 months), and 0/116 (0.0%) athletes had low BMD (Z-score <−1.0). No players had all three triad components (Fig. 1).

### Prevalence in the seven events for 116 athletes assigned to triad risk categories

In swimming, there were significantly more in the low-risk category than in the moderate and high-risk categories ($p = 0.007$). In athletics long-distance, there were significantly more in the moderate-risk category than in the low and high-risk categories ($p = 0.007$) (Table 1). In each scoring category, there was a high proportion (41/116, 35.3%) with a history of bone stress fracture, particularly in athletics long-distance (7/8, 87.5%) (Table 2).

### Number of injuries by triad risk categories (Table 3)

Since there was only one participant in the high-risk category, the high and moderate-risk categories were combined for the analysis. The number of injuries was 65 ($n = 41$) in one year. In the moderate and high-risk categories, there were significantly more in the injury group than in the non-injury group ($p = 0.01$). In the low-risk category, there were significantly more in the non-injury group than in the injury group ($p = 0.01$).

### Injured body part location and type of injury diagnosis by risk category

There was no significant difference in the injured body part location ($p = 0.713$). For stress fracture and bursitis, there were significantly more in the moderate and high-risk categories than in the low-risk category at injury diagnosis ($p = 0.014$) (Table 4).

## DISCUSSION

This study clarified the relationship between the triad risk assessment score and the one-year sports injury rate for female college students involved in multiple sports. To the best of our knowledge, there have been no studies of the relationship between the triad risk assessment score and the number of sports injuries.

Regarding the type of injury diagnosis by risk category, bone stress fracture and bursitis were significantly higher in the moderate and high-risk category than in the low-risk category. This result supported the hypothesis of this study. In previous studies, a higher number of triad risk factors was associated with an increased risk for bone stress injuries

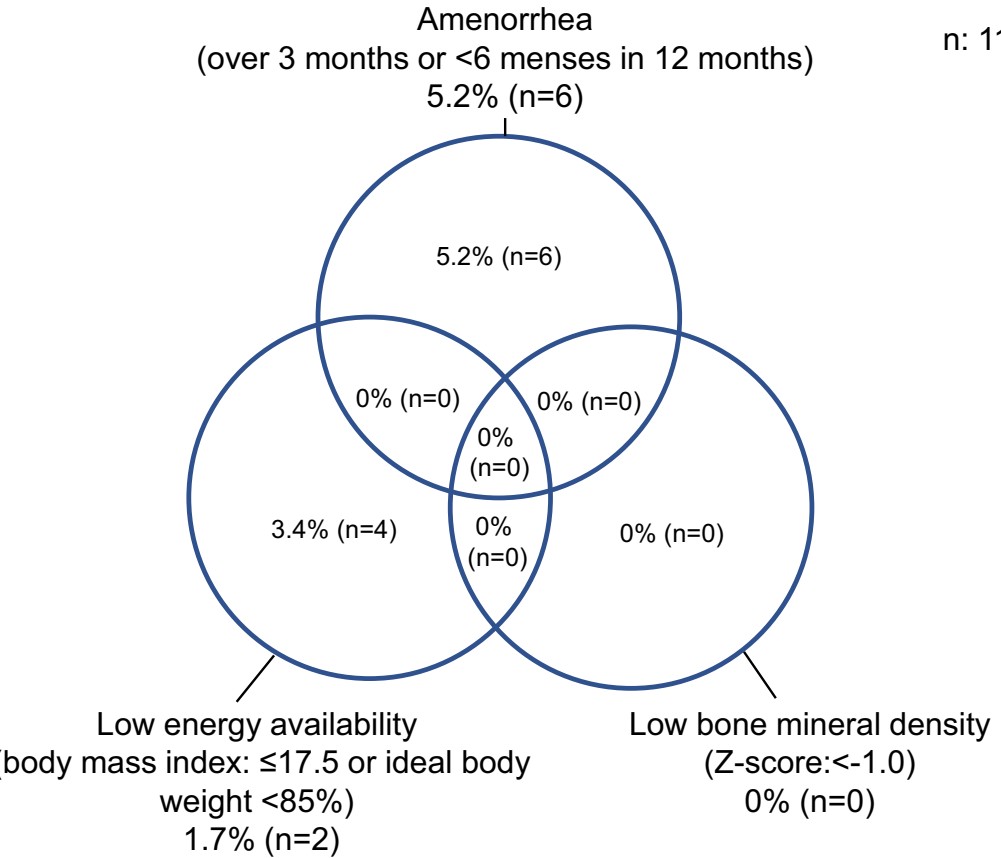

n: 116

**Figure 1 Percentage of athletes with the female athlete Triad.** A bone mineral density Z-score of < −1.0 in the heel is defined as low bone mineral density. The Triad is defined as energy intake minus energy expenditure of exercise relative to <30 kcal/kg of fat-free mass/d, but it is very difficult to calculate energy balance this way during medical examinations. Therefore, the American College of Sports Medicine defines low energy availability in adolescent athletes as a body weight <85% of ideal body weight, and for adult athletes in their 20s, a body mass index ≤17.5 kg/m². Therefore, these criteria were used in the present study. Body mass index was calculated as body weight (kg)/height (m²). To calculate ideal body weight, the formula recommended by The Japanese Society for Pediatric Endocrinology was used.

**Table 1 Prevalence in the 7 events for 116 athletes assigned to triad risk categories.**

| Sport | No. of athletes | Low risk | Moderate risk | High risk |
|---|---|---|---|---|
| Swimming | 11 | 11 (100.0)[a] | 0 (0.0) | 0 (0.0) |
| Athletics sprint | 19 | 11 (57.9) | 8 (42.1) | 0 (0.0) |
| Athletics long-distance | 8 | 2 (25.0) | 5 (62.5)[b] | 1 (12.5) |
| Athletics throwing/jumping | 8 | 7 (87.5) | 1 (12.5) | 0 (0.0) |
| Soccer | 27 | 19 (70.4) | 8 (29.6) | 0 (0.0) |
| Basketball | 26 | 21 (80.8) | 5 (19.2) | 0 (0.0) |
| Volleyball | 17 | 14 (82.4) | 3 (17.6) | 0 (0.0) |
| Total | 116 | 85 (73.3) | 30 (25.9) | 1 (0.8) |

Notes:
Data presented as $n$ (%).
[a] $p = 0.007$ vs. moderate and high-risk category.
[b] $p = 0.007$ vs. low and high-risk category.

Table 2 Number of athletes in each event by female athlete triad coalition scoring category.

| Category and risk score | Swimming (n = 11) | Athletic sprint (n = 19) | Athletic long-distance (n = 8) | Athletic throwing/ Jumping (n = 8) | Soccer (n = 27) | Basketball (n = 26) | Volleyball (n = 17) | Total (n = 116) |
|---|---|---|---|---|---|---|---|---|
| Low energy availability | | | | | | | | |
| Low score | 9 (7.8) | 16 (13.8 | 6 (5.2) | 8 (6.9) | 25 (21.6) | 25 (21.6) | 16 (13.8) | 105 (90.5) |
| Moderate score | 2 (1.7) | 3 (2.6) | 2 (1.7) | 0 (0.0) | 2 (1.7) | 1 (0.9) | 1 (0.9) | 11 (9.5) |
| High score | 0 (0.0) | 0 (0.0) | 0 (0.0) | 0 (0.0) | 0 (0.0) | 0 (0.0) | 0 (0.0) | 0 (0.0) |
| Body mass index or ideal body weight | | | | | | | | |
| Low score | 11 (9.5) | 14 (12.1) | 7 (6.0) | 7 (6.0) | 25 (21.6) | 25 (21.6) | 16 (13.8) | 105 (90.5) |
| Moderate score | 0 (0.0) | 3 (2.6) | 0 (0.0) | 1 (0.9) | 1 (0.9) | 1 (0.9) | 1 (0.9) | 7 (6.0) |
| High score | 0 (0.0) | 2 (1.7) | 1 (0.9) | 0 (0.0) | 1 (0.9) | 0 (0.0) | 0 (0.0) | 4 (3.4) |
| Age at menarche | | | | | | | | |
| Low score | 11 (9.5) | 17 (14.7) | 3 (2.6) | 7 (6.0) | 23 (19.8) | 21 (18.1) | 15 (12.9) | 97 (83.6) |
| Moderate score | 0 (0.0) | 2 (1.7) | 4 (3.4) | 1 (0.9) | 4 (3.4) | 4 (3.4) | 2 (1.7) | 17 (14.7) |
| High score | 0 (0.0) | 0 (0.0) | 1 (0.9) | 0 (0.0) | 0 (0.0) | 1 (0.9) | 0 (0.0) | 2 (1.7) |
| Oligomenorrhea/amenorrhea | | | | | | | | |
| Low score | 11 (9.5) | 13 (11.2) | 6 (5.2) | 7 (6.0) | 24 (20.7) | 21 (18.1) | 16 (13.8) | 98 (84.5) |
| Moderate score | 0 (0.0) | 5 (4.3) | 1 (0.9) | 1 (0.9) | 1 (0.9) | 2 (1.7) | 0 (0.0) | 10 (8.6) |
| High score | 0 (0.0) | 1 (0.9) | 1 (0.9) | 0 (0.0) | 2 (1.7) | 3 (2.6) | 1 (0.9) | 8 (6.9) |
| Low bone mineral density | | | | | | | | |
| Low score | 11 (9.5) | 19 (16.4) | 8 (6.9) | 8 (6.9) | 27 (23.3) | 26 (22.4) | 17 (14.7) | 116 (100.0) |
| Moderate score | 0 (0.0) | 0 (0.0) | 0 (0.0) | 0 (0.0) | 0 (0.0) | 0 (0.0) | 0 (0.0) | 0 (0.0) |
| High score | 0 (0.0) | 0 (0.0) | 0 (0.0) | 0 (0.0) | 0 (0.0) | 0 (0.0) | 0 (0.0) | 0 (0.0) |
| Stress reaction/fracture | | | | | | | | |
| Low score | 11 (9.5) | 10 (8.6) | 1 (0.9) | 6 (5.2) | 18 (15.5) | 20 (17.2) | 9 (7.8) | 75 (64.7) |
| Moderate score | 0 (0.0) | 8 (6.9) | 4 (3.4) | 2 (1.7) | 9 (7.8) | 6 (5.2) | 7 (6.0) | 36 (31.0) |
| High score | 0 (0.0) | 1 (0.9) | 3 (2.6) | 0 (0.0) | 0 (0.0) | 0 (0.0) | 1 (0.9) | 5 (4.3) |

Note:
Data presented as n (%).

Table 3 Number of injuries by triad risk category.

| Kind of sport | Injury group (n = 41) | | Non-injury group (n = 75) | |
|---|---|---|---|---|
| | Low risk (%) | Moderate and high risk (%) | Low risk (%) | Moderate and high risk (%) |
| Swimming | 3 (7.3) | 0 (0.0) | 8 (10.7) | 0 (0.0) |
| Athletics sprint | 0 (0.0) | 1 (2.4) | 13 (17.3) | 5 (6.7) |
| Athletics long-distance | 0 (0.0) | 3 (7.3) | 2 (2.7) | 3 (4.0) |
| Athletics throwing/jumping | 2 (4.9) | 1 (2.4) | 5 (6.7) | 0 (0.0) |
| Soccer | 12 (29.3) | 6 (14.6) | 8 (10.7) | 1 (1.3) |
| Basketball | 5 (12.2) | 2 (4.9) | 16 (21.3) | 3 (4.0) |
| Volleyball | 4 (9.8) | 2 (4.9) | 10 (13.3) | 1(1.3) |
| Total | 25 (61.0) | 16 (39.0)[a] | 62 (82.7)[b] | 13 (17.3) |

Notes:
Data presented as n (%).
[a] p = 0.01, Non-injury group with moderate risk (%).
[b] p = 0.01, Injury group with low risk (%).

**Table 4 Injury diagnosis by triad risk category.**

| Injury diagnosis | Low risk | Moderate and high risk |
|---|---|---|
| Concussion | 4 (6.2) | 1 (1.5) |
| Fracture | 2 (3.1) | 0 (0.0) |
| Stress fracture | 0 (0.0) | 4 (6.2)[a] |
| Other bone injuries | 0 (0.0) | 1 (1.5) |
| Dislocation, subluxation | 5 (7.7) | 1 (1.5) |
| Ligamentous rupture | 2 (3.1) | 0 (0.0) |
| Sprain | 8 (12.3) | 6 (9.2) |
| Lesion of meniscus or cartilage | 3 (4.6) | 0 (0.0) |
| Strain/muscle rupture/tear | 4 (6.2) | 0 (0.0) |
| Contusion/hematoma/bruise | 6 (9.2) | 0 (0.0) |
| Tendinosis/tendinopathy | 4 (6.2) | 2 (3.1) |
| Bursitis | 1 (1.5) | 4 (6.2)[b] |
| Muscle cramps or spasm | 1 (1.5) | 1 (1.5) |
| Nerve injury/spinal cord injury | 1 (1.5) | 0 (0.0) |
| Others (nail trouble, heatstroke) | 3 (4.6) | 1 (1.5) |
| Total | 44 (67.7) | 21 (32.3) |

Notes:
Data presented as *n* (%).
Only injuries that occurred are listed.
[a] $p = 0.014$ vs. low-risk category for stress fracture.
[b] $p = 0.014$ vs. low-risk category for bursitis.

and low BMD (*Barrack et al., 2014*; *Gibbs et al., 2014*; *Tenforde et al., 2013*). Furthermore, for athletes with component of the triad, the risk of developing bone stress fractures was about 3–5 times higher than that of athletes with no components of the triad (*Mallinson & De Souza, 2014*). Therefore, this study was considered to have supported the results of the previous studies. However, it is necessary to examine bursitis in greater detail in the future.

In the present study, there were 4/116 (3.4%) athletes with LEA with or without DE/ED, 6/116 (5.2%) with amenorrhea, and 0/116 (0.0%) with low BMD. No athletes had all three triad components. In previous study of elite Japanese athletes, the number of athletes with LEA was 42/300 (14.0%), with amenorrhea was 117/300 (39.0%), and with low BMD was 68/300 (22.7%). Seventeen athletes (5.7%) had both amenorrhea and LEA, whereas 39 (13%) had both amenorrhea and low BMD, and two (0.7%) had low BMD and LEA. Sixteen (5.3%) had all three components of the triad (*Nose-Ogura et al., 2019*). In previous study of American collegiate athletes, the number of athletes with LEA was 2/323 (0.6%), the number with oligomenorrhea or amenorrhea was 64/239 (26.8%), and the number with low BMD was 19/323 (5.9%) (*Tenforde et al., 2017*). The cause for the differences may be related to the differences in competition level and measurement methods.

In swimming, the number in the low-risk category was significantly higher than in the moderate and high-risk categories. In athletics long-distance, the number in the moderate-risk category was significantly higher than in the low-risk category. In a previous

study, athletics (64/86; 74.4%) (*Nose-Ogura et al., 2019*), track (0/4; 0.0%) (*Tenforde et al., 2017*), cycling (3/4; 75.0%) (*Nose-Ogura et al., 2019*), swimming (7/11; 63.6%) (*Nose-Ogura et al., 2019*), gymnastics (7/7; 100.0%) (*Nose-Ogura et al., 2019*) (9/16; 56.2%) (*Tenforde et al., 2017*), rhythmic gymnastics (31/35; 88.6%) (*Nose-Ogura et al., 2019*), and cross-country (23/47; 48.9%) (*Tenforde et al., 2017*) were in the moderate or high-risk categories. Although there is no clear consensus, it was considered that there were many endurance and esthetic sports athletes in the middle- and high-risk categories.

In addition, for each scoring category, there was a large proportion (41/116, 35.3%) with a history of bone stress fractures, particularly in athletics long-distance (7/8; 87.8%). In previous studies, female athletes were at a higher risk of bone stress fractures than male athletes (*De Souza et al., 2014*; *Nose-Ogura et al., 2019*). It has also been reported that the frequency of bone stress fractures among 1,616 female Japanese athletes and 537 controls (non-athletes) was 22.6% for athletes competing at the international level, 23.3% for athletes competing at the national level, 20.8% for athletes competing at the local level, 18.8% for athletes competing at other levels, and 4.3% for controls (*Takamatsu & Kitawaki, 2016*). Therefore, the athletes in the present study had a high rate of bone stress fractures. Furthermore, careful consideration should be given to the reason why significantly more athletes were in the moderate-risk category than in the low-risk category in athletics long-distance.

Several limitations must be considered in this study. First, the 1,000 athlete exposures could not be calculated. Second, the survey of injuries during sports activities was conducted for one season from April 2018 to March 2019, but medical examinations and anthropometry were conducted from August 2018 to January 2019. Therefore, this was not a prospective study. In the future, prospective research will be needed. Third, the actual LEA, defined as energy intake minus energy expenditure of exercise relative to fat-free mass <30 kcal/kg of FFM/d, was not measured in this study. Fourth, BMD was measured on the heel of the right leg using an ultrasonic bone densitometer, and calculated Z-score in this study. BDM from ultrasonic bone densitometer cannot be used for diagnostic classification and it is not clinically useful for monitoring the effects of therapy (*Lewiecki, Richmond & Miller, 2006*). And a calcaneus ultrasound T-score is commonly higher than a central X-ray absorptiometry T-score and could give the false impression (*Lewiecki & Lane, 2008*). However, Z-score, which is not used with the WHO classification, rather than T-scores, are preferred for reporting the results for premenopausal women, men under the age of 50 years and children (*Lewiecki & Lane, 2008*).

## CONCLUSIONS

In this study, regarding the number of injured athletes by risk category, in the moderate and high-risk categories, there were significantly more athletes in the injury group than in the non-injury group. In addition, there were significantly more athletes in the moderate and high-risk categories than in the low-risk category with bone stress fractures and bursitis. These results suggest that athletes in the moderate and high-risk categories of the Triad may be at increased risk of injury.

## ACKNOWLEDGEMENTS

The authors would like to acknowledge and thank all female college athletes and coaches.

### Funding

The authors received no funding for this work.

### Competing Interests

The authors declare that they have no competing interests.

### Author Contributions

- Mutsuaki Edama conceived and designed the experiments, performed the experiments, analyzed the data, prepared figures and/or tables, authored or reviewed drafts of the paper, and approved the final draft.
- Hiromi Inaba performed the experiments, analyzed the data, prepared figures and/or tables, authored or reviewed drafts of the paper, and approved the final draft.
- Fumi Hoshino performed the experiments, prepared figures and/or tables, authored or reviewed drafts of the paper, and approved the final draft.
- Saya Natsui analyzed the data, prepared figures and/or tables, and approved the final draft.
- Sae Maruyama analyzed the data, authored or reviewed drafts of the paper, and approved the final draft.
- Go Omori conceived and designed the experiments, authored or reviewed drafts of the paper, and approved the final draft.

### Human Ethics

The following information was supplied relating to ethical approvals (i.e., approving body and any reference numbers):

The Niigata University of Health and Welfare of Ethical approval to carry out the study within its facilities (18032).

### Data Availability

Raw measurements are available as a Supplemental File.

### Supplemental Information

Supplemental information for this article can be found online at http://dx.doi.org/10.7717/peerj.11092#supplemental-information.

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
