# Peer review of "The relationship between the female athlete triad and injury rates in collegiate female athletes"

_PeerJ, doi:10.7717/peerj.11092_

## Round 0.1 · original submission · Major Revisions

Dear Dr. Edama,

Two reviewers have examined your manuscript and provided detailed commentary. I have reviewed the comments and would like to invite you to revise and resubmit your manuscript, and respond to reviewer comments with point-by-point responses or rebuttals. If you do modify your manuscript, please reference the exact location of the changes within the manuscript to allow reviewers to more easily see the revisions.

Of note, your approach to determining low energy availability is of major concern to reviewers. This requires further justification and perhaps a different approach. Both reviewers made detailed comments about this.

Also note that both reviewers provided annotated manuscripts as part of their reviews.

Please note that an invitation to revise your manuscript does not guarantee eventual acceptance, and any revisions you choose to submit will be subject to additional peer review.

Best Wishes
Zachary Zenko

·

Basic reporting

The manuscript should be reviewed for consistency, tense and grammar.
The references are complete and appropriate. The introduction does not build a case for the purpose and hypothesis. Both the Triad and REDs were introduced but there was no discussion of their differences or similarities. The main purpose was to use the Female Athlete Triad Cumulative Risk Assessment but previous uses of the tool were not included and the reason to use this tool was not clear. What are the instances of injury in female athletes? Why would this tool help?
In lines 60-62; It seems that RED-S is a well established theoretical concept, when in fact it’s the opposite. Based on the DeSouza 2014b reference that they provided, which doesn’t (currently) support the use of RED-S as a well-established model, should include some a disclaimer or limitations as to how the IOC actually misinterprets or is lacking evidence behind their claims.
Raw data was shared in accordance with the Data Sharing policy.
The titles of the tables are incorrect for Tables Consistent terminology should be used for the tool Female Athlete Triad Cumulative Risk Assessment. The tables refer to the measure as female athlete triad coalition scoring category and triad risk categories, consistent terms are needed.

Experimental design

The main hypothesis of this work is interesting but is not supported by the current introduction. The background of why this is important is unclear and how the current data will contribute to the knowledge gap. The Introduction needs to be re-worked so the readers understand “why” this is an important topic and how these data will add to the knowledge gap in this area.
Medical Examinations: Who completed the examinations and attaching the questionnaire would be helpful. Was injury self report?
Anthropometry: Insufficient detail on all measures was given.
The ultrasound methods and estimation of energy deficiency should each have their own sections to explain how the data was collected. It is unclear what measures were collected from the ultrasound device and how BMD was determined from the standard measures of Speed of Sound (SOS) and broadband ultrasound attenuation (BUA) that are collected from these methods (REFS).
Ln 136 more information is needed on how the Z Scores from ultrasound measures were determined. Does the Female Athlete Triad Cumulative Risk Assessment allow for measures of BMD other than from DXA measurements?
Ln 137 was the history of stress fracture self report or verified through medical records? How was a stress fracture vs a trabecular stress fracture determined?
Ln 115-117 The triad is not defined by the energy availability but one factor of the triad is low energy availability. More detail on the classifications in De Souza et al should be included as well as references to other studies that have utilized this method to determine energy availability. Furthermore, the limitations of this approach should be addressed.
Ln 118-119: low EA is not entirely defined as BMI <17.5 kg/m^2. Athletes may have low EA but a ‘normal’ BMI. Looking at other methods such as detailed food intake and energy expenditure can be used to determine this (DeSouza 2014a).
Injury Rate:
Were the injury data collected based on medical reports or by team/athlete physical therapists during the season? How many physical therapists were reporting the data and how was consistency between therapists determined and verified?
Statistical analysis:
More detail on the statistical analysis is needed. Which comparisons are testing the hypothesis from the introduction, “ the moderate and high-risk 93 groups have higher injury rates than the low-risk group on the triad risk assessment score”. Also which statistical program was used.

Validity of the findings

Results:
Ln 157 Patients Characteristics, should not be referred to as patients but as participants or athletes. Could information on their age or duration of sports participation be added to these results.
Ln 167 Title of this section does not clarify the results that are being addressed. Can the exact p values be included in the text or table not just the reference to p < 0.05.
Ln 162: These data should be from parts of Table 2 but it is unclear where these numbers are from.
Ln 171-172: indicate that a high history of bone stress fracture was from the combined moderate and high risk. Also reporting on the other factors of the triad may be interesting from table 2.
Ln 175 add “one” to the “since there was only one participant…”
Ln 178-179 please report exact p values for these 2 comparisons.
Ln 181-184 Is Table 4 needed and more detail on what Table 5 is illustrating.
Discussion:
The first paragraph of the discussion should highlight the main results on the data. Was the hypothesis supported? What was the main take away from these data. The second paragraph of the discussion should be in the results section. Paragraph 5 addresses the primary hypothesis and should be addressed earlier in the discussion. The limitations say that injury rate could not be determined but the first paragraph states “there have been no studies of the relationship between the triad risk assessment 190 score and sports injury rate.” Terminology and the use of injury risk and injury rate need to be defined and clarified throughout the manuscript.
The conclusion states: “This study clarified the relationship between the triad risk assessment score and the one year sports injury rate for female college students of multiple sports.” It is not clear how the relationship has been clarified with the data presented. The discussion needs to convince the reader of this claim and in its current format the conclusion does not seem to be supported. In addition, the conclusion states “ This was not a prospective survey, but its 255 results suggest that athletes with RED-S may be at increased risk of injury. “ The authors used the Triad Cumulative Risk assessment so why is the conclusion focused on REDs? Both should be introduced and discussed in the manuscript, but should not be used interchangeably throughout the manuscript.

Additional comments

I am attaching a pdf with additional comments (footnotes) that can be. found on page 36.

Reviewer 2 ·

Basic reporting

Basic reporting is satisfactory, however I would recommend further review of the literature with regards to classification and diagnostics of low energy availability and RED-S vs. female athlete triad.

Experimental design

Thank you for your submission. I find your broadening the scope of injury surveillance beyond bone stress injury interesting. I do however have strong methodological concerns regarding your choice of BMI as a sole marker of energy availability. I worry that this may influence the results of your paper. I would recommend further explanation in your methods regarding injury diagnosis.

Validity of the findings

Please see above statement.

Also please consider your statement in Line 200 that your athletes were "well managed". This statement is ambiguous and may be misinterpreted if not better explained.

Additional comments

Please see comments attached

Annotated reviews are not available for download in order to protect the identity of reviewers who chose to remain anonymous.

---

## Round 0.2 · Minor Revisions

Two expert reviewers (including one new reviewer) provided their comments on your manuscript. Both reviewers acknowledged its improvement and I concur. Please provide a point-by-point response and rebuttal so that your revision can be more easily processed.

·

Basic reporting

For the Low BMD category:
In manuscript:
For low BMD, athletes with a Z139 score ≤−2 were scored 2, and those between −1 and −2 were scored 1; a score of 0 was given to
140 those over −1

But in Table 2: the low category is actually "good BMD"? At first glance it looks like there are a majority of athletes with low BMD. Maybe the title of the column could reiterate that Risk is (Low, Moderate, High).

The methods for the US need to be elaborated. How was BMD derived from SOS or BUA measurements. Is there a good correlation between US and DXA BMD measures?

Experimental design

No comment

Validity of the findings

no comment

Additional comments

My main concern is the use of BMD from and US measure. There seems to be some controversy in the literature and more detail in the Methods is needed and a comment in the limitations. https://www.ncbi.nlm.nih.gov/pmc/articles/PMC3891842/#:~:text=A%20number%20of%20devices%20are,and%20quantitative%20CT%20(QCT).

·

Basic reporting

Basic Reporting

The authors appear to have addressed the previous reviewer comments to a good standard, there are a few issues within the introduction that would be useful to be addressed prior to publication.

The raw data is provided, although some coding descriptions, and replacing X/O with 0/1 would perhaps be beneficial to allow for other researchers to quickly perform counts of the data.

L64. You introduce “many studies” here that examine bone stress fractures and amenorrhea but fail to provide the references to these many studies. Could the authors include some evidence here (either a meta-analysis or some other high quality sources)?

L.67
You describe here a relationship between the menstrual cycle and injury risk, but I would suggest that this is unlikely and realistically the situation is more complex and nuanced than this. A higher injury rate in female athletes is likely to be due to numerous factors and to simplify this down to menstrual cycle is too simple. Some more clarification on this would be useful here.

Experimental design

The main hypothesis of this work is interesting as stated by previous reviewers, and the reworking of the introduction does provide this with more context in the literature.

The investigation is sound, however I have some concerns about the data analysis and the pre-registration of this study. Was this study pre-registered? This is a requirement of research involving human participants and if this study was not pre-registered could the authors provide a justification as to why and I urge them to consider this approach in future.

The data presented here is quite superficial given what has been collected, and presented within the raw data file. Some descriptive statistics about the population studied would be useful, as this would allow those who may wish to understand how the Triad may play a role in athletes that they work with (or themselves) to establish if this is a population similar to the one they work with. This would also allow the reader to understand this study in comparison to the previous literature.

I would concur with previous reviewer on the value of Table 4, as I am unsure what it brings to the study, especially as this data is replicated in Table 5.. This information is presented within the raw data file, and feel the authors could refer to this in the manuscript as opposed to including within the manuscript.

Validity of the findings

The conclusions drawn from this study look robust, and the combined with the request of the manuscript [and on conjunction with the previously requested amendments] have no concerns regarding this manuscript beyond those outlined above.

Additional comments

No comment

---

## Round 0.3 · accepted · Accept

Thank you for productively engaging with the peer-review process and addressing multiple reviewer concerns.

Three expert reviewers have read your manuscript and provided feedback, and, after two rounds of revision, your manuscript is improved. I am now recommending acceptance.